# Major Plant in Herbal Mixture Gan-Mai-Da-Zao for the Alleviation of Depression in Rat Models

**DOI:** 10.3390/plants11030258

**Published:** 2022-01-19

**Authors:** Ying-Xiao Li, Kai-Chun Cheng, Chao-Tien Hsu, Juei-Tang Cheng, Ting-Ting Yang

**Affiliations:** 1Department of Nursing, Tzu Chi University of Science and Technology, Hualien 970302, Taiwan; yxli@ems.tcust.edu.tw; 2Department of Pharmacy, College of Pharmacy and Health Care, Tajen University, Pingtung 90741, Taiwan; kc-cheng@tajen.edu.tw; 3Department of Pathology, E-Da Hospital, I-Shou University, Kaohsiung 82445, Taiwan; ed103797@edah.org.tw; 4Department of Medical Research, Chi-Mei Medical Center, Tainan City 71004, Taiwan; 5School of Chinese Medicine for Post-Baccalaureate, I-Shou University, Kaohsiung 82445, Taiwan

**Keywords:** Gan-Mai-Da-Zao, depression, brain-derived neurotrophic factor, serotonin transporter, unpredictable chronic mild stress

## Abstract

Gan-Mai-Da-Zao (GMDZ) is a well-known product in Chinese traditional medicine and includes three major plants: blighted wheat (Fu Mai), licorice (Gan Cao), and jujube (Da Zao). GMDZ is widely used as an efficacious and well-tolerated prescription for depression in clinics. The present study was designed to investigate the main plant of GMDZ for its antidepressant-like effect using the unpredictable chronic mild stress (UCMS) model on rats who received an injection with p-chlorophenylalanine (PCPA) to produce the chemical model. In rats subjected to the UCMS model, forced swim tests, open field tests, and sucrose preference tests were applied to estimate the chronic effect of GMDZ. We found that the oral administration of GMDZ for 21 days significantly alleviated the behavior in rats with depression induced by either UCMS or PCPA. The expression levels of the serotonin transporter (5-HTT) and brain-derived neurotrophic factor (BDNF) in the hippocampus of the rats with depression were markedly increased by GMDZ. Additionally, rats that received the herbal mixture without licorice showed a markedly lower response than GMDZ. These results suggest that GMDZ may alleviate the depressive-like behaviors in depressive rats, possibly via licorice (Gan Cao), to increase 5-HTT and BDNF signals in the hippocampus. The present study confirmed the antidepressant-like effects of GMDZ. Additionally, licorice (Gan Cao) may play a key role in the effectiveness of GMDZ.

## 1. Introduction

Major depression is a common psychiatric disorder and it may lead to emotional depression, suicidal tendencies, and a recurrence of morbidity [1]. Dysfunction of the serotonin (5-HT) system is commonly considered to be the cause of depression. Depression has been shown as the negative factor that affects the rate of adult hippocampal neurogenesis. The serotonin transporter (5-HTT) responsible for the reuptake of 5-HT is related to the role of 5-HT in neurodevelopmental processes [2]. The evidence from animal models and human studies indicates that reduced function of 5-HTT is associated with the decreased expression of brain-derived neurotrophic factor (BDNF) [3]. Additionally, BDNF is highly expressed in the adult hippocampus and hypothalamus [4], and is involved in the etiopathology of mood disorders [5]. In depressive patients, serum BDNF levels are markedly decreased [6] that can be restored by antidepressant treatment [7]. Although today’s treatments for depression have greatly improved, it is still necessary to find more safe and effective agents to prevent depression.

Traditional Chinese medicine (TCM) has shown the therapeutic effects of depression [8,9]. GMDZ is one of the well-known products in TCM and it has widely been used to treat depressive patients in Asia. Despite the large variety of TCM patterns among participants, dozens of herbal formulas for depression were GMDZ-based [10]. This was first documented in the Chinese medical book Jin-Gui-Yao-Lue (Synopsis of Prescriptions of the Golden Chamber) written by Dr. Zongjing Zhang (AD 152-219) [11]. GMDZ is believed to be effective for depression [12]. Clinical studies indicated that GMDZ decoction is an efficacious and well-tolerated antidepressant prescription for depressive disorders, even postpartum depression [13,14,15]. Moreover, GMDZ could protect hippocampal neurons against glutamate toxicity in depression-like rats [16,17]. The composition of GMDZ includes three major plants: blighted wheat (FuMai, M), licorice (GanCao, G), and jujube (DaZao, D) [16]. The combination of three plants may enhance the efficiency and/or reduce toxicity in clinical applications. However, the role of these components in GMDZ is still vague.

To investigate the antidepressant-like effects of GMDZ, unpredictable chronic mild stress (UCMS) was used in the present study. The behavior tests, including a forced swimming test (FST), open field test (OFT), and sucrose preference test (SPT), were then performed. Moreover, we compared the variations between one herb-deleted mixture and GMDZ to understand the main plant in GMDZ using the rats with depression induced by pretreatment with a serotonin synthesis inhibitor, p-chlorophenylalanine (PCPA). The levels of 5-HTT and BDNF in the hippocampus of rats were also assessed to obtain further insight into the mechanism(s) regarding the antidepressive effects of GMDZ.

## 2. Results

### 2.1. Chronic GMDZ Treatment Ameliorated Depression-Like Behaviors in UCMS Rats and PCPA Treated Rats

FST shows a high predictive validity for antidepressant activity. OFT is classically used to assess anxiety in rodents. In the present study, the UCMS group significantly prolonged the immobility time in the FST (Figure 1a); and also reduced the time spent at the center and decreased the total distance traveled in the OFT compared with the control group (Figure 1b,c). Similar results were observed in depressive rats induced by PCPA in FST (Figure 1e) and OFT (Figure 1f,g). To assess the antidepressant-like effects of GMDZ, rats were orally administrated GMDZ for 21 days while fluoxetine (10 mg/kg) was included as the positive control. The results showed that GMDZ produced a significant reduction in the duration of immobility as well as the time in the center and distance of traveling in OFT. It indicated that GMDZ significantly ameliorated depression-like behaviors compared with the vehicle-treated animals. Additionally, fluoxetine showed marked effects on FST and OFT in the UCMS group which were not observed in the PCPA-induced model.

The SPT is a procedure which is used to measure the hedonic value of sucrose, typically reduced in animals with depression-like disorders. Both the UCMS and PCPA groups (Figure 1h) showed a significant decrease in sucrose consumption as compared with the control. However, the sucrose consumption in both models was significantly restored by the chronic administration of GMDZ. PCPA-induced stressed rats treated with fluoxetine did not differ in sucrose preference from the vehicle-treated group. Therefore, GMDZ produces antidepressant effects through different mechanisms to fluoxetine.

### 2.2. Effects of Two-Herb Mixture of GMDZ on Behavioral Tests in PCPA-Induced Rats

To understand the major plant in GMDZ, a one-plant deletion from the mixture was prepared. Then, five groups of PCPA-induced rats were administered with GMDZ, G and M, M and DZ, G and DZ, and the vehicle, respectively, for 21 days. In both G and M and G and DZ groups, rats showed a shorter immobility time in FST (Figure 2a), a longer central zone duration and a greater total distance in OFT (Figure 2b,c), and a higher sucrose intake in SPT (Figure 2d) which ameliorated the depression-like behaviors. However, M and DZ did not show an obvious effect in FST, OFT, or SPT, respectively. These results suggest that G seems to play a crucial role in the effect of GMDZ using PCPA-induced rats showing depression-like behavior.

### 2.3. Chronic GMDZ Treatment Restores the 5-HTT and BDNF Levels in the Hippocampus of PCPA Treated Rats

To investigate the potential mechanism(s) underlying GMDZ-induced antidepressant effects, we determined the gene expressions and protein levels of 5-HTT (Figure 3a,b) and BDNF in the hippocampus of PCPA-treated rats (Figure 3c,d). The mRNA and protein levels of 5-HTT were significantly decreased in the PCPA-treated group, which was restored by GMDZ. Additionally, chronic treatment with GMDZ also increased the protein and mRNA levels of BDNF in the hippocampus of PCPA-treated rats.

Same as the results in behavior experiments, G and M or G and DZ treatment significantly restored the mRNA and protein levels of 5-HTT. Moreover, as shown in Figure 3, G and M or G and DZ treatment also restored the protein level and mRNA level of BDNF. However, the treatment of M and DZ did not modify the expressions of 5-HTT and BDNF in the hippocampus. It supports that G played an important role in GMDZ for depression improvement.

### 2.4. Effects of Glycyrrhizic Acid on 5-HTT and BDNF Expression in the Corticosterone-Treated H19-7 Cell Line

We hypothesized that licorice played a major role in GMDZ mixture, since the antidepressant-like effect in G and M and G and DZ treatments seemed to be more significant than that in the M and DZ treatment group. In addition, long-term exposure to stress or high glucocorticoid levels leads to depression-like behavior in rodents [18]. Therefore, we investigated the potential mechanism of glycyrrhizic acid (the active component of licorice) for corticosterone-induced stress injury in cells. Our results show that the in vitro findings correlate with the vivo results. It showed that chronic exposure of H19-7 cells to corticosterone markedly decreased the gene expressions of 5-HTT (Figure 4a) and BDNF (Figure 4b). Interestingly, glycyrrhizic acid significantly reversed these expressions in a dose-dependent manner.

## 3. Discussion

UCMS rats exhibited a significantly increased immobility time in the FST, decreased locomotor activity in OFT, and reduced sucrose intake in the SPT, as described previously [19]. In the present study, we demonstrated that chronic administration of GMDZ ameliorated depression-like behaviors in UCMS rats. Moreover, depletion of 5-HT by PCPA failed to block the antidepressant action of GMDZ, indicating that the action mechanisms of GMDZ varied from fluoxetine which is one of SSRI. Otherwise, we found that GMDZ may promote the BDNF signaling pathway and enhance 5-HTT expression in the hippocampus in the depressive rats.

Serotonin (5-HT) released from the axon terminalis is selectively taken up from the synaptic cleft into these terminals via the 5-HTT [2]. The decrease of 5-HTT expressions is associated with stress-induced anxiety and depression-like behaviors [20]. It demonstrates an interaction between 5-HT and BDNF [21]. The loss of BDNF appears to exacerbate neurochemical and behavioral abnormalities in 5-HTT mutant mice [22]. BDNF modulated the 5-HTT gene promoter to influence the function of 5-HTT [23]. Our results are consistent with previous findings that 5-HTT-gene and protein expressions were suppressed in PCPA rats [16]. Administration of GMDZ reversed the expression level of 5-HTT compared with the vehicle-treated UCMS group. Additionally, chronic GMDZ administration the stress-induced reversed the decrease of BDNF level [24] in the hippocampus. Therefore, GMDZ may play an important role in the 5-HTT and BDNF regulation. GMDZ promoted the increase of 5-HTT and BDNF expressions, which may contribute to the antidepressant effect [7].

The effects of GMDZ may include improved cerebral microcirculatory regulation, mood stabilization, and the alleviation of impatience, as noted in a previous report [15]. As the herbal mixture of GMDZ seems highly complex, the present study aimed to understand the main plant that played a major role in the therapeutic effects of GMDZ. Using the deletion of one plant from the original mixture in GMDZ, three products (G and M, G and DZ, and M and DZ) were administrated to PCPA-treated rats, respectively. This showed that G and M and G and DZ treatments, but not M and DZ treatment, produced significant antidepressant-like effects in the PCPA-induced model, same as the effects of GMDZ. Additionally, G and M or G and DZ administration also reversed the decreased 5-HTT and BDNF levels in depression. As the antidepressant-like effect in the G and M and G and DZ treatments seemed more significant than that in the M and DZ treatment group, licorice (G) might play a major role in the effectiveness of GMDZ. It has been documented that licorice induces an antidepressant-like effect in animals [25,26]. GanCao (licorice) contains triterpenoid glycosides and flavonoid glycosides. Glycyrrhizin or glycyrrhizic acid as the active ingredient in licorice has been extensively studied [27]. Glycyrrhizic acid has been reported to show anti-inflammatory and anti-nociceptive activities in mice [28]. The present study found that glycyrrhizic acid increases the 5-HTT and BDNF expressions, both were reduced in H19-7 cells treated with corticosterone [23]. Therefore, glycyrrhizic acid as one of the active principles in GanCao (licorice) seems to have participated in the alleviation of depressive disorders in animals. However, the real action mechanism(s) shall be clarified in the future.

According to the traditional TCM theory, the blighted wheat seems to have a predominant function in GMDZ. It seems possible that the active constituent from blighted wheat is through the conversion of gut microbiota in animals. Stress leads to anxiety/depression by altering the gut microbiota, and it is possible to improve anxiety and depression by probiotics modulation [29]. The blighted wheat seems beneficial for the protection of exhaustive physical exercise, oxidative stress injury on brain tissues [30]. However, it needs more investigations for the blighted wheat in the future. Otherwise, Dazao contains various triterpenoids (e.g., betulinic acid and oleanolic acid) and glycosides [31]. But it seems only plays a supporting role in the regulation of depression. However, the systemic administration of betulinic acid and the oral administration of oleanolic acid show analgesic effects on acetic acid-induced writhing in the animal model [32,33].

## 4. Materials and Methods

### 4.1. Preparation of Extracts of the GMDZ Decoction

GMDZ is mainly prepared from three dried raw plants: licorice (Glycyrrhiza uralensis fisch, GanCao, G), blighted wheat (Triticum aestivum L, FuMai, M), and jujuba (Ziziphus jujuba Mill, DaZao, DZ). In the present study, the used GMDZ was a commercial product purchased from Sun Ten Pharmaceutical Co., Ltd. (Taipei, Taiwan). The specification of the GMDZ preparation is as follows: Glycyrrhiza uralensis (root and rhizome): 18.75%; Triticum aestivum (fruit): 62.50%; Ziziphus jujuba (fruit): 18.75%. After extraction in distilled water (ratio 1:10), the product of GMDZ was similar to that in a previous report [34]. To identify the major plant in GMDZ, the commercial product of each plant in GMDZ was also purchased. Then, we used the single deletion method to mimic the original prescription and new mixtures in three, such as GanCao + FuMai (G and M), FuMai + DaZao (M and DZ), GanCao + DaZao (G and DZ), were obtained. Each product has the same ratio as that in GMDZ decoction. The amount of each deleted plant was replaced by the starch to reach the same amount of GMDZ. Then, the treated doses were expressed as the dried weight of each product per bodyweight of the animals (g/kg body weight).

### 4.2. Experimental Animals

Male Sprague-Dawley rats (250–300 g) obtained from National Animal Center (Taipei, Taiwan) were maintained in the animal center of Chi-Mei Medical Center (Tainan, Taiwan). In brief, they were housed in pairs on a 12/12-hr light/dark cycle (light beginning at 7:00 am)) with ad libitum access to food and water except during behavioral tests. The project was approved by the Institutional Animal Care and Use Committee of Chi-Mei Medical Center (No. 105122622). All the animal procedures were performed according to the Guide for the Care and Use of Laboratory Animals published by the US National Institutes of Health (NIH Publication No. 85-23, revised 1996).

### 4.3. Experimental Design

To investigate the antidepressant-like effect of GMDZ, rats were randomly divided into four groups (eight rats in each group): control group, UCMS model group, UCMS + fluoxetine group, and UCMS + GMDZ group. Additionally, to find the major herb, rats induced by PCPA were also used as PCPA + fluoxetine group, PCPA + GMDZ group, PCPA + G and M group, PCPA + M and DZ group, and PCPA + G and DZ group.

PCPA (Sigma, St. Louis, MO, USA), a specific inhibitor of serotonin (5-HT) biosynthesis, was administered (100 mg/kg) once a day for seven days, as described previously [35]. PCPA has shown a high degree of 5-HT depletion (>90%) yielded by similar treatment [36]. On the seventh day, changes in 5-HTT expression in the hippocampus were confirmed by Western blots. Two days after the model induced by PCPA, rats were administered with GMDZ (2.5 g/kg) [34], G and M mixture (2.5 g/kg), M and D mixture (2.5 g/kg), G and D mixture (2.5 g/kg) and fluoxetine (10 mg/kg) once daily by oral gavage for a three-week period.

### 4.4. UCMS Procedure

According to the previous report [37], the UCMS-induced depressive animal model was induced. Experimental rats (*n* = 8 per group) were exposed to unpredictable mild stressors randomly every day in four weeks. The stressors applied included the following: physical restraint (1h), 1 min tail pinch (2.5 cm from the end of the tail), reversed light/dark cycle (24 h), overnight illumination (12 h), soiled cage (12 h), and cage tilt (18 h, 45°). Each stressor was randomly assigned two or three times over the experimental period. The non-stressed control rats were normally housed in groups (three to four per cage) in the other room, and the stressed rats were singly housed [37]. At least 12 h of rest was provided between a stressor and a test to avoid effects of acute stress [38].

After the first week, the animals were administered with GMDZ (2.5 g/kg) or fluoxetine (10 mg/kg) by oral gavage once a day for three weeks. The dosage was applied according to a previous study [34]. Behavioral tests were performed 2 h after the last treatment.

### 4.5. Behavioral Tests

The sequence of the behavioral test was SPT, OPT, and FST. There was a three-day time interval between these tests.

SPT: To evaluate the anhedonia response [39], rats were exposed to two identical bottles (one containing tap water and the other 1% sucrose solution) for 1 h, followed by 12 h of tap water and food deprivation [40]. The bottles were weighed before and after the 1 h test period; the sucrose preference (%) was then determined. Animals were habituated three days to the two-bottle choice (both bottles were filled with tap water and placed through the top of the cage lid) before the test.

OPT: To measure locomotion and anxious behaviors, rats were placed in an open field area made of a 70 × 70 × 40 cm wooden box and equipped with an infrared floor to measure locomotor activity [41]. The arena was subdivided into a central and a peripheral zone. Rats were placed in the open field boxes for 5 min under normal light conditions, and the locomotor activity and time stay in central was tracked with a video system (Viewpoint, Lyon, France). Individual animals were gently placed in the same corner of the apparatus in all trials.

FST: To assess learned-helplessness, rats were gently placed in a clear plastic cylinder (height = 40 cm; diameter = 30 cm) filled with water to 30 cm high at 24 °C ± 0.5 °C for 6 min [42]. Immobility time was calculated by subtracting active time from the total time. The behavior experiments were recorded using a side-mounted camera and assessed using a video tracking software.

### 4.6. Tissue Preparation

Rats were sacrificed 24 h after the behavior experiments. The treatments continued during these days. Rats were euthanized by intraperitoneal injection (IP) of a lethal dose of pentobarbital. The brain was quickly removed and immediately frozen in liquid nitrogen for further analysis.

### 4.7. Cell Cultures

Rat-derived hippocampus H19-7 cell line cells (CRL-2526; American Type Culture Collection, Manassas, VA, USA) were maintained at 37 °C and 5% CO_2_ in Dulbecco’s modified Eagle’s medium (DMEM; HyClone, South Logan, UT, USA) with 4 mM l-glutamine that was adjusted with sodium bicarbonate (1.5 g/L), glucose (4.5 g/L), G418 (200 μg/mL), and puromycin (1 μg/mL) and supplemented with 10% fetal bovine serum [43]. Cells (1 × 10^6^) were plated on 60-mm culture dishes, and at 80% confluence. H19-7 cells were incubated with or without the rat stress hormone, corticosterone, at the concentration of 1 μM, in differentiating medium for five days, as described previously [44]. Then, glycyrrhizic acid at various concentrations was pretreated with the corticosterone-incubated H19-7 cells for 24 h. Finally, the H19-7 cells were collected for assay as described below.

### 4.8. Western Blotting Analysis

The protein concentration of the total protein lysates was determined using the BCA protein assay kit (Pierce Biotechnology, Rockford, IL, USA). The protein lysates (30 µg) were separated by electrophoresis and transferred to a polyvinylidene difluoride membrane (Millipore, Billerica, MA, USA). After blocking and washings, the following primary antibodies were incubated at 4 °C overnight: anti-BDNF (1:1000; Abcam, Cambridge, UK), anti-5-HTT (1:1000; Millipore, Billerica, MA, USA), anti-β-actin (1:5000, Sigma-Aldrich, St. Louis, MO, USA). The protein bands were visualized using the enhanced chemiluminescence kit (PerkinElmer, Boston, MA, USA). The quantification was determined using software (Gel-Pro Analyzer version 4.0 software (Media Cybernetics Inc., Silver Spring, MD, USA).

### 4.9. Real-Time Reverse Transcription-Polymerase Chain Reaction

According to our previous method [45], the mRNA expression levels of each signal were determined. In brief, total RNA was extracted from the cell lysates with TRIzol reagent (Carlsbad, CA, USA). Total RNA (200 ng) was reverse-transcribed into cDNA with random hexamer primers (Roche Diagnostics, Mannheim, Germany). All PCR experiments were performed using a LightCycler (Roche Diagnostics GmbH, Mannheim, Germany). The concentration of each PCR product was calculated relative to a corresponding standard curve. The relative gene expression was subsequently indicated as the ratio of the target gene level to that of β-actin. The primers for BDNF, 5-HTT and β-actin are listed as follows:BDNF F: 5′-GCAGTCAAGTGCCTTTGGAG-3′;BDNF R: 5′-CGGCATCCAGGTAATTTTTG -3′;5-HTT F: 5′-CATCAGCCCTCTGTTTCTCC -3′;5-HTT R: 5′-CGGACGACATCCCTATGC -3′;β-actin F: 5′-CTAAGGCCAACCGTGAAAAG-3′;β-actin R: 5′-GCCTGGATGGCTACGTACA-3′.

### 4.10. Statistical Analysis

Data are presented as the mean ± standard errors (SE). Statistical analysis was conducted by using SPSS Version 21. Comparisons between groups were performed using the one-way analysis of variance (ANOVA) with Bonferroni’s posthoc method. The *p* values of less than 0.05 were considered statistically significant.

## 5. Conclusions

The present study demonstrated that GMDZ ameliorated depression-like behaviors in rats. GMDZ regulated the 5-HTT and BDNF expression in the depression model. In addition, GanCao (licorice) played a crucial function in GMDZ decoction for antidepression.

## Figures and Tables

**Figure 1 plants-11-00258-f001:**
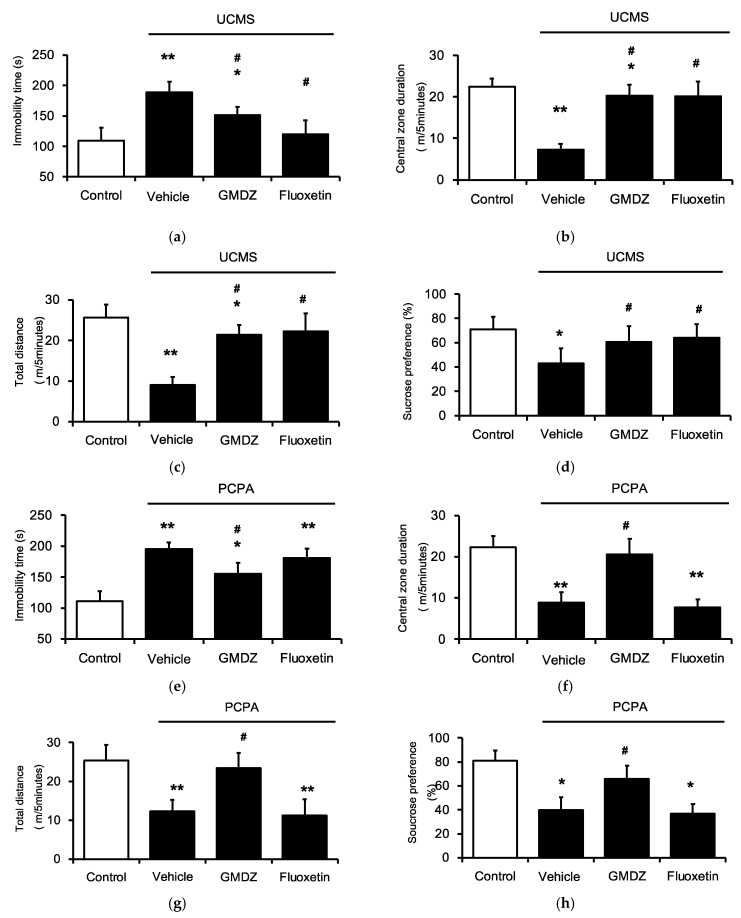
GMDZ ameliorated depression-like behaviors in two rat models induced by the UCMS or PCPA injection. (**a**) The changes of immobility time in FST in UCMS model; (**b**) the changes of time in central in OFT in UCMS model; (**c**) the changes of traveling distance in OFT in UCMS model; (**d**) the changes of sucrose solution consumption (%) in SPT in UCMS model; (**e**) the changes of immobility time in FST in PCPA treated groups; (**f**) the changes of time in central in OFT in PCPA treated groups; (**g**) the changes of traveling distance in OFT in PCPA treated groups; (**h**) the changes of sucrose solution consumption (%) in SPT in PCPA treated groups. Responses to fluoxetine used as the positive control. Data are expressed as mean ± SE (*n* = 8). * *p* < 0.05, ** *p* < 0.01 compared with the normal control group; # *p* < 0.05 compared with vehicle-treated group.

**Figure 2 plants-11-00258-f002:**
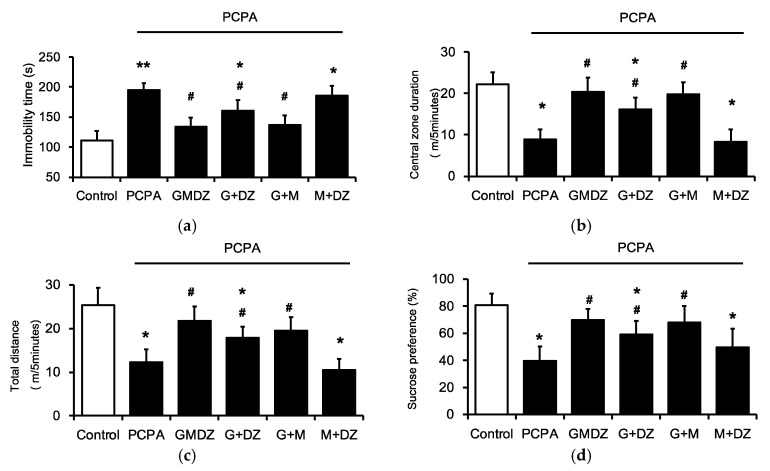
Effects of the one herb-deleted mixture of GMDZ in depression-like behaviors using the PCPA treated rats. (**a**) The changes of immobility time in FST; (**b**) the changes of time in central in OFT; (**c**) the changes of traveling distance in OFT; (**d**) the changes of sucrose solution consumption (%) in SPT. Data are expressed as mean ± SE (*n* = 8). * *p* < 0.05, ** *p* < 0.01 compared with the normal control; # *p* < 0.05 compared with vehicle-treated group.

**Figure 3 plants-11-00258-f003:**
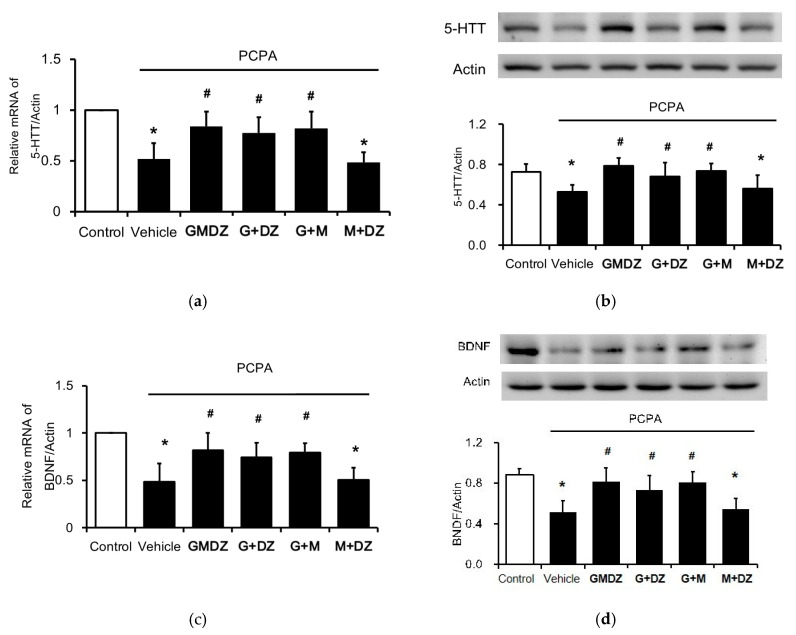
Effects of GMDZ and the one plant-deleted mixture of GMDZ on the expressions of 5-HTT and BDNF in the hippocampus. (**a**) The mRNA levels of 5-HTT; (**b**) The protein levels of 5-HTT; (**c**) the mRNA levels of BDNF; (**d**) the protein levels of BDNF. Data are expressed as mean ± SE (*n* = 8). * *p* < 0.05 compared with the normal control; # *p* < 0.05 compared with vehicle-treated group.

**Figure 4 plants-11-00258-f004:**
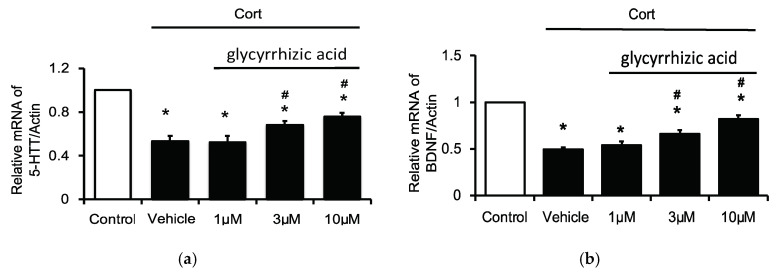
Effects of glycyrrhizic acid (GA) on 5-HTT and BDNF expressions in the corticosterone-treated H19-7 cells. H19-7 cells were incubated under normal differentiating conditions in the presence or absence of corticosterone (Cort) at concentration of 1 μM for five days. Then, cells were respectively incubated with glycyrrhizic acid at 1 μM, 3 μM, or 10 μM for 24 h. (**a**) The mRNA levels of 5-HTT; (**b**) the mRNA levels of BDNF. Data are expressed as mean ± SE (*n* = 8). * *p* < 0.05 compared to the cells without treatment of corticosterone (control). # *p* < 0.05 compared with the vehicle-treated group.

## Data Availability

The data is confidentiality.

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
