# Peer review of "Major Plant in Herbal Mixture Gan-Mai-Da-Zao for the Alleviation of Depression in Rat Models"

_plants, 2022, doi:10.3390/plants11030258_

Round 1
Reviewer 1 Report
This article describe a preclinical examineation of a tradicional Chinese medicine, which is a mixture, with the hope 1) to reveal some mechanism; 2) to identify the major, most effective component. Despite its importance I cannot accept the ms in its present form, as the aim is not clearly stated and now the experiments seems to be also rather random.
In fact the UCMS model is just used to confirm clinical data (?) while a more acute model is used to study both research aims. The in vitro experiment was out of the blue.
Specific comments:
Abstract: " In the development of multiple plants" ??? is it a mixture of plant extracts?
" The present study is designed to find out the major plant" ??? The aim is not clear and the model used is also confounding in the abstract as well as in the introduction.
A major conclusion on a single component, however, only two component together were examined in each case. Why not a single component was used?
Fig.1 as UCMS or PCPA is at the top of each subfigure, at the bottom I would suggest to add Veh as vehicle instead of UCMS or PCPA.
The statistical analysis is missing throughout. (F and p values, degree of freedom; posthoc comparison?)
2.3. the title suggests that here also UCMS model was used, but Fig3 contains only PCPA model. Thus, no data or statistic is presented here.
Suddenly cort treatment also appeared on a cell line... (in fact chronic cort can be used in vivo as well)
Fig. 4 bootom line: I do not understand the name of the columns. Cort here means Veh treatment and the number means glycyrrhizic acid? If so, indicate it on the figure.
The discussion is overwritten, contains many too heuristic sentences like "There-174 fore, licorice (G) plays a major role in the antidepression-like effect of GMDZ." I do not really see it, as it was never examined alone!
How was the behaviour analysed? (online with a stopwatch)?
Behavioural analysis should have been done during the dark active phase. When was teh lights on?
4.6. After the tests, but when exactly? What was the sequence and ITI for the behavioural tests?
Author Response
Reply to Comments (plants-1496546)
We appreciate the time and effort taken by all the reviewers to review our manuscript, as well as their constructive comments and suggestions made in order to improve its quality. We have carefully addressed all issues raised and made changes to the manuscript accordingly. All changes in the manuscript are marked as red text.
Reviewer #1
This article describes a preclinical examination of a traditional Chinese medicine, which is a mixture, with the hope 1) to reveal some mechanism; 2) to identify the major, most effective component. Despite its importance I cannot accept the ms in its present form, as the aim is not clearly stated and now the experiments seem to be also rather random.
In fact, the UCMS model is just used to confirm clinical data. while a more acute model is used to study both research aims. The in vitro experiment was out of the blue.
Reply:
Thanks for your insightful comment. In the in vitro experiments, we confirmed our hypothesis that licorice plays a key role in the GMDZ mixture. Therefore, the effect of glycyrrhizic acid (the active component of licorice) on corticosterone-induced stress injury in the cell was investigated. We have included this information in the Results section. Thank you very much.
Specific comments:
Abstract: " In the development of multiple plants" ??? is it a mixture of plant extracts?
Reply: Thanks for your helpful indications. Gan-Mai-Da-Zao (GMDZ) is a mixture of plant extracts, which includes three major plants: wheat (Fu Mai), licorice (Gan Cao), and jujube (Da Zao). Thank you so much.
" The present study is designed to find out the major plant”??? The aim is not clear and the model used is also confounding in the abstract as well as in the introduction.
Reply: Following your suggestions, we have modified the Abstract in the revised version, as shown in the highlights. Thank you very much.
A major conclusion on a single component, however, only two components together were examined in each case. Why not a single component was used?
Reply: Thank you for your valuable comment. In this new version, we have improved the conclusion part of the Abstract, as shown in the highlights.
Fig.1 as UCMS or PCPA is at the top of each subfigure, at the bottom I would suggest to add Veh as vehicle instead of UCMS or PCPA.
Reply: Thank you for the helpful comments. We have revised Fig.1 according to your suggestion.
2.3. the title suggests that here also UCMS model was used, but Fig3 contains only PCPA model. Thus, no data or statistic is presented here.
Reply: Thank you for raising this concern. We have revised the descriptions of Fig. 3 following your suggestion.
Suddenly cort treatment also appeared on a cell line... (in fact chronic cort can be used in vivo as well)
Reply: Thank you for the valuable indication. In the revision, we have added the purpose of the cort experiment into the Results section, as shown in the highlights.
Fig. 4 bottom line: I do not understand the name of the columns. Cort here means Veh treatment and the number means glycyrrhizic acid? If so, indicate it on the figure.
Reply: Thank you for the helpful comments. This has been adjusted in Fig. 4 in accordance with your suggestion.
The discussion is overwritten, contains many too heuristic sentences like "There-174 fore, licorice (G) plays a major role in the antidepression-like effect of GMDZ." I do not really see it, as it was never examined alone!
Reply: Thank you for raising this concern. We have rewritten the Discussion section following this suggestion.
How was the behavior analyzed? (Online with a stopwatch)?
Reply: Thank you for this suggestion. The behavior experiments were recorded by a camera inserted into the bath. We have improved the Methods section following this suggestion.
Behavioral analysis should have been done during the dark active phase. When was the lights on?
Reply: Thank you for raising this concern. In the present experiment, light began at 7:00 am. We have added this into the Methods section.
4.6. After the tests, but when exactly? What was the sequence and ITI for the behavioral tests?
Reply: Thank you for the comments. We have added the details of behavior experiments into the Methods section, as shown in the highlights.

Reviewer 2 Report
The article under review (Major plant in herbal mixture Gan-Mai-Da-Zao for the alleviation of depression in rat models) describes the effects observed in experimentally stressed rats after the administration of a Traditional Chinese Medicine as a plant decoction. Furthermore, the importance of each plant in the mixture was assessed.
The introduction provides a concise background.
Regarding materials and Methods, the nature of the tested preparation (Traditional Chinese Medicine GMDZ) should be explained better in order to allow experimental reproducibility of the publication. Detail the plant part for the three species used to prepare GMDZ (root, aerial parts, seeds?), as well as the exact preparation of GMDZ (time of extraction).
The authors use standard research methods for the field of antidepressant medicines. Results are clear and well illustrated.
In the discussion section, comment on the ratio between the dosage used in the lab experiments (2.5 g GMDZ/kg body weight) and the dose that is usually prescribed to humans in Traditional Chinese Medicine.
While English style and grammar of the body text are standard, the abstract has several English style problems. Please check it by a native speaker.
The conclusions are supported by experimental evidence, are short and concise.
Author Response
Reply to Comments (plants-1496546)
We appreciate the time and effort taken by all the reviewers to review our manuscript, as well as their constructive comments and suggestions made in order to improve its quality. We have carefully addressed all issues raised and made changes to the manuscript accordingly. All changes in the manuscript are marked as red text.
Reviewer #2
The article under review (Major plant in herbal mixture Gan-Mai-Da-Zao for the alleviation of depression in rat models) describes the effects observed in experimentally stressed rats after the administration of a Traditional Chinese Medicine as a plant decoction. Furthermore, the importance of each plant in the mixture was assessed.
The introduction provides a concise background.
Regarding materials and Methods, the nature of the tested preparation (Traditional Chinese Medicine GMDZ) should be explained better in order to allow experimental reproducibility of the publication. Detail the plant part for the three species used to prepare GMDZ (root, aerial parts, seeds?), as well as the exact preparation of GMDZ (time of extraction).
Reply: Thank you for your insightful comments. We have included the information of the GMDZ mixture into the Methods section. “GMDZ: Glycyrrhiza uralensis (root and rhizome) 18.75% Triticum aestivum (fruit) 62.50% Ziziphus jujuba (fruit) 18.75%.” In the present study, the used GMDZ was a commercial product purchased from Sun Ten Pharmaceutical Co., Ltd (Taipei, Taiwan).
The authors use standard research methods for the field of antidepressant medicines. Results are clear and well-illustrated.
In the discussion section, comment on the ratio between the dosage used in the lab experiments (2.5 g GMDZ/kg body weight) and the dose that is usually prescribed to humans in Traditional Chinese Medicine.
Reply: In preliminary experiments, we compared the antidepressant-like effects of GMDZ at various doses. GMDZ at 2.5 g/kg body weight showed the best result. We appreciate your helpful suggestions.
While English style and grammar of the body text are standard, the abstract has several English style problems. Please check it by a native speaker.
Reply: The MDPI English Editing team has checked the revised version. Please find the certificate below. Thank you very much.
The conclusions are supported by experimental evidence, are short and concise.

Reviewer 3 Report
In this manuscript, the authors concluded that licorice is the major herbal in the Gan-Mai-Da-Zao decoction and stated that glycyrrhizic acid is the active ingredient in licorice for treating depression. According to this manuscript, however, treating depression by using licorice only instead of Gan-Mai-Da-Zao makes more meaningful to the audients since the authors stated both Wheat and Dazao makes no contribution in this decoction for treating depression. So far, the function of licorice in treating depression has been widely studied, and the possible pathways and the active ingredients/fractions of it have received lots of attraction. Hence, this manuscript is lack of novelty and did not explain or solve any mechanism-related problem regarding Gan-Mai-Da-Zao.
- Line 81, ‘both the UCMS group (Figure 1d) and PCPA group (Figure 1h) showed a significant decrease in sucrose consumption as com-pared with the control’ . The Y-axis label and unit of figure 1d is Total distance and m/5minutes, respectively, which are wrong. Please correct them.
- Figures 1c and 1d are identical, please either correct the error or explain the figures.
- Line 84-85, ‘The increase of sucrose consumption by GMDZ was near to that in the fluoxetine-treated group in the UCMS group.’ There is no evidence representing the sucrose consumption of the UCMS group. Please add the data, which support the authors’ statement.
- Line 85-86, ‘As described above, fluoxetine failed to recover the sucrose preference in the PCPA-induced group’, there is no indication before this sentence mentioned the activity of fluoxetine on sucrose preference test, please add the corresponding information or rephrase the sentence.
- Line 87, ‘fluoxetine failed to recover the sucrose preference in the PCPA-induced group’, it is unable to make this conclusion since the sucrose consumption of the UCMS group missing.
- Figure 1 caption, a semicolon is missed between (g) and (h)
- Line 101-103, ‘In the new prepared mixture, G & M or G & DZ treatment ameliorated the depression-like behaviors in FST (Figure 2a), 102 OFT (Figure 2b, 2c) and SPT (Figure 2d),’ it is hard for audients to understand the statement, please rephrase this sentence by presenting the data one by one to avoid misunderstanding.
- Line 104-105, ‘Moreover, the antidepressant effects of G & M and G & DZ were same as GMDZ group’, this is no evidence that can support this evidence according to this abovementioned information, please add the evidence.
- Please correct the title of section 2-4 to ‘Effects of glycyrrhizic acid on 5-HTT and BDNF expression in the corticosterone-treated H19- 131 7 cell line’.
- Section 2-4, Please rephrase this section to let the audients understand the purpose of learning the activity of glycyrhizic acid on 5-HTT and BDNF expression since glycyrhizic acid has no yet been introduced in the above sections.
Author Response
Reply to Comments (plants-1496546)
We appreciate the time and effort taken by all the reviewers to review our manuscript, as well as their constructive comments and suggestions made in order to improve its quality. We have carefully addressed all issues raised and made changes to the manuscript accordingly. All changes in the manuscript are marked as red text.
Reviewer #3
In this manuscript, the authors concluded that licorice is the major herbal in the Gan-Mai-Da-Zao decoction and stated that glycyrrhizic acid is the active ingredient in licorice for treating depression. According to this manuscript, however, treating depression by using licorice only instead of Gan-Mai-Da-Zao makes more meaningful to the audients since the authors stated both Wheat and Dazao makes no contribution in this decoction for treating depression. So far, the function of licorice in treating depression has been widely studied, and the possible pathways and the active ingredients/fractions of it have received lots of attraction. Hence, this manuscript is lack of novelty and did not explain or solve any mechanism-related problem regarding Gan-Mai-Da-Zao.
Line 81, ‘both the UCMS group (Figure 1d) and PCPA group (Figure 1h) showed a significant decrease in sucrose consumption as com-pared with the control’. The Y-axis label and unit of figure 1d is Total distance and m/5minutes, respectively, which are wrong. Please correct them.
Reply: Thank you for pointing this out. We have revised this part in the Results section. Thank you very much.
Figures 1c and 1d are identical, please either correct the error or explain the figures.
Reply: Thank you, and we apologize for the oversight. It has been corrected in this revision. Many thanks, again.
Line 84-85, ‘The increase of sucrose consumption by GMDZ was near to that in the fluoxetine-treated group in the UCMS group.’ There is no evidence representing the sucrose consumption of the UCMS group. Please add the data, which support the authors’ statement.
Reply: Thank you for your helpful comments. Fig.1 has been revised accordingly.
Line 85-86, ‘As described above, fluoxetine failed to recover the sucrose preference in the PCPA-induced group’, there is no indication before this sentence mentioned the activity of fluoxetine on sucrose preference test, please add the corresponding information or rephrase the sentence.
Reply: Thank you for your comments. We have improved this part in the revision.
Line 87, ‘fluoxetine failed to recover the sucrose preference in the PCPA-induced group’, it is unable to make this conclusion since the sucrose consumption of the UCMS group missing.
Reply: Thanks for raising this concern. We have rewritten the results following your instructions.
Figure 1 caption, a semicolon is missed between (g) and (h)
Reply: Thanks for the kind reminder. We have added a semicolon between Fig 1g and Fig 1h.
Line 101-103, ‘In the new prepared mixture, G & M or G & DZ treatment ameliorated the depression-like behaviors in FST (Figure 2a), 102 OFT (Figure 2b, 2c) and SPT (Figure 2d),’ it is hard for audients to understand the statement, please rephrase this sentence by presenting the data one by one to avoid misunderstanding.
Reply: Thank you for your helpful comments. We have improved the results accordingly.
Line 104-105, ‘Moreover, the antidepressant effects of G & M and G & DZ were same as GMDZ group’, this is no evidence that can support this evidence according to this abovementioned information, please add the evidence.
Reply: Thanks for your helpful suggestion. We have improved the Results section accordingly.
Please correct the title of section 2-4 to ‘Effects of glycyrrhizic acid on 5-HTT and BDNF expression in the corticosterone-treated H19- 131 7 cell line’.
Reply: Thanks for the kind reminder. We have improved the title of Section 2-4 according to your suggestions.
Section 2-4, Please rephrase this section to let the audients understand the purpose of learning the activity of glycyrhizic acid on 5-HTT and BDNF expression since glycyrhizic acid has no yet been introduced in the above sections.
Reply: Thanks for your helpful suggestion. We have improved the Results section 2-4 following your instructions. Thank you very much.

Round 2
Reviewer 1 Report
Although the ms imporved a lot, there are still many thing, which needs further clarification.
I still not understand, that why not a single compound alone was investigated rather a mixture of two and conclusion is drawn on the missing one.
Specific comments:
L 61 delete " rats received"
L 66 In which model, after which treatment (single-prolonged) were these parameters examined? How it is related to the previously mentioned two models?
L72 OFT has nothing to do with depression, reflect - if something- anxiety.
L74 replace "which" with "and"; delete "in"
L83 replace "but it" with "which"
L133 It is still out of the blue. Add some parts to the introduction mentioning in vitro experiment
L159 "nervous terminal"??? axon terminalis perhaps
L166 As I see these parameters were studied in the PCPA and not CMS model.
L170 Helpful for antidepressants? Perhaps may contribute to the antidepressant effect.
Please explain why fluoxetine was not effective here.
L173-180 Shorten it avoiding pure repetition of the results. (e.g. state only that only mixture containing licorice was effective in alleviating depressive-like behavioural and molecular alterations)
L194 verb is missing
L246 for one week
L 246 vs 253? Was it one or three weeks?
L254 Was the CMS continued during treatment?
L261 How was a camera withing the bath? We normally record it from above or side and analyse later. How was the recording analysed? When was it done? 12h after the last stress adn 2h after last treatment?
L262 When was this test done? After FST? Or on a separate group of rats?
L 269 Was there a habituation to two bottles?
L273 Were the treatments continued during these days?
Author Response
Reply to Comments (plants-1496546-R2)
We appreciate the time and effort taken by you to review our revised version. We have further carefully addressed all issues raised and made changes to the manuscript accordingly. All changes in the new version are marked as red text. Although you have declined to review it, we still like to answer your questions in below.
Reviewer #1
Although the ms imporved a lot, there are still many thing, which needs further clarification.
I still not understand, that why not a single compound alone was investigated rather a mixture of two and conclusion is drawn on the missing one.
Reply: It is followed the main targets of this special issue. Thank you very much.
Specific comments:
L 61 delete " rats received". It has been done.
L 66 In which model, after which treatment (single-prolonged) were these parameters examined? How it is related to the previously mentioned two models?
Reply: Sorry, we have no enough time to search the model because we were asked to complete the revision within three days only.
L72 OFT has nothing to do with depression, reflect - if something- anxiety.
Reply: Maybe you are right although it needs to be clarified in advance.
L74 replace "which" with "and"; delete "in". It has been done.
L83 replace "but it" with "which". It has been done.
L133 It is still out of the blue. Add some parts to the introduction mentioning in vitro experiment. It has been done.
L159 "nervous terminal"??? axon terminalis perhaps. It has been revised.
L166 As I see these parameters were studied in the PCPA and not CMS model.
It has been revised.
L170 Helpful for antidepressants? Perhaps may contribute to the antidepressant effect.
Reply: Thanks, and it needs more investigations in advance.
Please explain why fluoxetine was not effective here.
Reply: Fluoxetine is widely used as reference. However, we are used the models in screening of same product only.
L173-180 Shorten it avoiding pure repetition of the results. (e.g. state only that only mixture containing licorice was effective in alleviating depressive-like behavioural and molecular alterations). It has been done.
L194 verb is missing. It has been done.
L246 for one week. It has been done.
L 246 vs 253? Was it one or three weeks? It has been done.
L254 Was the CMS continued during treatment? Yes.
L261 How was a camera withing the bath? We normally record it from above or side and analyse later. How was the recording analysed? When was it done? 12h after the last stress adn 2h after last treatment? Thanks, it has been rephrased.
L262 When was this test done? After FST? Or on a separate group of rats? Thanks, and it has been revised.
L 269 Was there a habituation to two bottles? Yes, it has been added.
L273 Were the treatments continued during these days? Yes, it has been indicated. Thank you very much.

Reviewer 3 Report
The authors have addressed all my concerns.
Author Response
Reply to Comments (plants-1496546-R2)
Thanks for your time and effort to review our revision. We have improved the manuscript accordingly.
Reviewer #3
English language and style are fine/minor spell check required.
Reply: The MDPI English Editing team has checked the new version. Please find the certificate below. Thank you very much.
